# Design of 1D Photonic Crystals Sustaining Optical Surface Modes

Valery Konopsky

Institute of Spectroscopy, Fizicheskaya 5, 108840 Moscow, Russia; konopsky@isan.troitsk.ru

**Abstract:** An impedance approach has been implemented to design truncated 1D photonic crystals, sustaining optical surface modes, with any predetermined wavelength and wavevector. The implementation is realized as a free Windows program that calculates both the thicknesses of the double layers and the thickness of the final truncated layer at given refractive indices of the layers. The dispersion of the refractive indices can be given in the form of the Sellmeier/Drude formulas or in the form of a wavelength-n-k table. For mixed layers, the Maxwell Garnett theory can be used. This approach is suitable for studying and visualizing the field distribution inside photonic crystals, dispersion, and other aspects of the designed structures that sustain optical surface modes. Therefore, this program should promote scientific development and implementation of practical applications in this area.

**Keywords:** thin films; multilayers; optical surface modes; photonic crystal waveguides; optical bloch surface waves; long-range surface plasmons

## 1. Introduction

Optical surface waves are excitations of electromagnetic modes that exist near the interface between two media. Photonic Crystal Surface Modes (PC SMs), which are also called 'photonic band-gap surface modes', 'modes of (asymmetric) planar Bragg waveguide', 'surface waves in periodic layered medium', 'photonic crystal surface waves', 'optical Bloch surface waves', and 'surface waves in multilayer coating' are modes that are bound to the external surface of an one-dimensional (1D) photonic crystal.

Photonic crystals are materials that possess a periodic modulation of their refraction indices on the scale of the wavelength of light [1]. Such materials can exhibit photonic band gaps that are very much like the electronic band gaps for electron waves traveling in the periodic potential of the crystal. In both cases, frequency intervals exist in which wave propagation is forbidden. This analogy may be extended [2] to include surface levels, which can exist in band gaps of electronic crystals. In PCs, they correspond to optical surface modes with dispersion curves located inside the photonic band gap.

The one-dimensional photonic crystal (1D PC) is a simple periodic multi-layer stack. Optical surface modes in 1D PCs were studied in the 1970s, both theoretically [3,4] and experimentally [5]. Twenty years later, the excitation of optical surface modes in a Kretschmann-like configuration was demonstrated [6,7]. A scheme of the Kretschmann-like excitation of PC SMs is presented in Figure 1. In recent years, the PC SMs have been used in ever-widening applications in the fields of optical sensors [8–14], optical biosensors [15–23] and in other fields [24–30].

To excite the PC SM at any predetermined wavelength and at any predetermined wavevector (i.e., at any predetermined angle in the Kretschmann scheme), it is necessary to calculate the thickness of the last dielectric layer, which depends on the thicknesses of the double layers and their refractive indices (RIs), as well as RI of external environment. The thickness of the double layer should be pre-calculated in advance in order to maximize a

photonic band gap at these predetermined wavelength and wavevector, since the confinement of the PC SM near the interface is the result of the photonic band gap on one side and the total internal reflection (TIR) on another side of the interface.

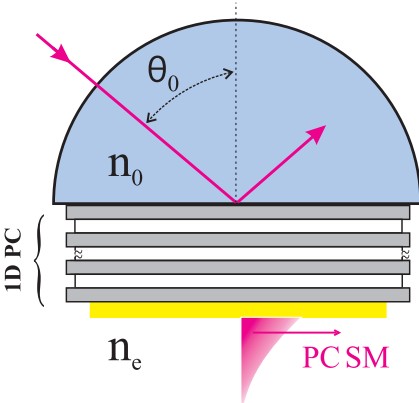

**Figure 1.** Excitation of optical surface waves in a Kretschmann-like scheme.

Various approaches and strategies have been proposed to find the optimal parameters of the 1D PC structure and the thickness of the terminated layer [31–34]. Here, to solve this problem, we present a free Windows program [35] based on an impedance approach, which was used in [36,37]. This software permits both modeling and visualization of parameters of an 1D PC structure before submitting it for fabrication.

## 2. Materials and Methods

### 2.1. Theoretical Background: Impedance Approach

All calculations in the program are performed using the impedance approach, which provides a unified theoretical description of multilayer systems for s- and p-polarizations by the same equations. With this approach, the 'normal impedance' $Z$ is the ratio of the tangential components of the electric field to the magnetic field, which has the following forms (in a $j$th layer) for s- and p-polarizations [36,38,39]:

$$Z_{s(j)} = \frac{1}{n_j \cos(\theta_j)} = \frac{1}{n_j \sqrt{1 - (\rho/n_j)^2}} \tag{1}$$

$$Z_{p(j)} = \frac{\cos(\theta_j)}{n_j} = \frac{\sqrt{1 - (\rho/n_j)^2}}{n_j}. \tag{2}$$

Here and hereafter the numerical aperture $\rho = n \sin(\theta)$ will be used as an angle variable in a planar multilayer system. It is a unified angle variable for all layers, since according to Snell's law $\rho = n_j \sin(\theta_j)$, for any $j$.

### 2.2. Reflection and Transmission for a Multilayer with N Layers

One of the advantages of using the impedance approach is the ability to represent the impedances of *several* layers as a *single* 'apparent' input impedance through a recursive relation. For example, if the multilayer is made up of $N$ plane-parallel, homogeneous, isotropic layers (with refractive indices $n_j$ and geometrical thicknesses $d_j$, where $j = 1, 2, \ldots, N$) between semi-infinite incident $_{(0)}$ and external $_{(e)}$ media (see Figure 2), the apparent input impedance $Z_{(j)}^{\text{into}}$ of a semi-infinite external medium $_{(e)}$ and layers from $N$ to $j$ may be calculated by the following *recursion relation* [36–39]:

$$Z_{(j)}^{\text{into}} = Z_{(j)} \frac{Z_{(j+1)}^{\text{into}} - iZ_{(j)} \tan(\alpha_j)}{Z_{(j)} - iZ_{(j+1)}^{\text{into}} \tan(\alpha_j)}, \tag{3}$$

where $\alpha_j = k_{z(j)}d_j = (2\pi/\lambda)n_j\cos(\theta_j)d_j$; $j = N, N-1, \ldots, 2, 1$ and $Z^{\text{into}}_{(N+1)} = Z_{(N+1)} = Z_{(e)}$, $n_{N+1} = n_e$ while $d_{N+1} = d_e = 0$ by definition.

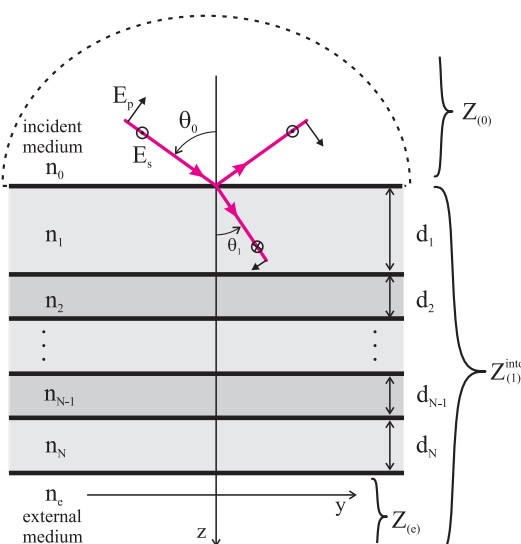

**Figure 2.** Reflection and transmission for a multilayer with $N$ layers in terms of impedance.

This procedure should be continued recursively from the last layer (where, on the first iteration, $Z_{(e)}$ would be replaced by $Z^{\text{into}}_{(N)}$) – to the first layer until $Z^{\text{into}}_{(1)}$ is obtained. The equation for reflection coefficients of *s*- or *p*-polarized waves from any complex multilayer (see Figure 2) has a very simple form in the impedance terms:

$$r = \frac{Z^{\text{into}}_{(1)} - Z_{(0)}}{Z^{\text{into}}_{(1)} + Z_{(0)}}, \tag{4}$$

where $Z^{\text{into}}_{(1)}$ is an apparent input impedance for a multilayer, i.e., it is the impedance that is seen by an incoming wave as it approaches to the interface. In the absence of the multilayer, it reduces to the standard Fresnel's formula for *s*- or *p*-polarization, respectively.

Fresnel's formulas for multilayer transmission coefficients are as follows:

$$t_s = \prod_{j=0}^{j=N} t_{s\left(^{j+1}_{\phantom{j}j}\right)}, \quad \text{with}$$

$$t_{s\left(^{j+1}_{\phantom{j}j}\right)} = -\frac{\left(Z^{\text{into}}_{s(j+1)} + Z_{s(j+1)}\right)}{\left(Z^{\text{into}}_{s(j+1)} + Z_{s(j)}\right)} e^{i\alpha_{j+1}} \tag{5}$$

and:

$$t_p = \prod_{j=0}^{j=N} t_{p\left(^{j+1}_{\phantom{j}j}\right)}, \quad \text{with}$$

$$t_{p\left(^{j+1}_{\phantom{j}j}\right)} = -\frac{n_j Z_{p(j)}}{n_{j+1} Z_{p(j+1)}} \frac{\left(Z^{\text{into}}_{p(j+1)} + Z_{p(j+1)}\right)}{\left(Z^{\text{into}}_{p(j+1)} + Z_{p(j)}\right)} e^{i\alpha_{j+1}} \tag{6}$$

where $t_{\left(^{j+1}_{\phantom{j}j}\right)}$ are transmission coefficients at an interface between the $j$ layer and the $j+1$ layer.

### 2.3. Input Impedance of a SEMI-Infinite 1D PC

A strategy for finding multilayer thicknesses and the thickness of the last, truncated layer that sustain the propagation of optical surface waves, is still a matter of discussion [32]. Our strategy in the first step is to choose a double layer thickness of the PC multilayer structure to maximize the bandgap extinction at a pre-selected wavelength and wavevector, as this ensures maximum confinement of the PC SM from the PC side of the interface. Then we solve the dispersion equation PC SM to find the thickness of the truncated (last) dielectric layer.

Both operations employ the input impedances of a semi-infinite 1D PC and then apply the obtained thicknesses to a practical finite 1D PC. This approach shows an excellent match of the resonance parameters with a pre-selected wavelength and wavevector. The impedance of a semi-infinite 1D PC was derived in [36]:

$$Z_{(PC)}^{\text{into}} = -\frac{i}{2} \frac{\left( (Z_{(2)}^2 - Z_{(1)}^2) \tan(\alpha_2) \tan(\alpha_1) \pm \sqrt{s} \right)}{Z_{(2)} \tan(\alpha_1) + Z_{(1)} \tan(\alpha_2)}, \tag{7}$$

where:

$$s = -4Z_{(1)}Z_{(2)}\left( Z_{(2)} \tan(\alpha_1) + Z_{(1)} \tan(\alpha_2) \right)\left( Z_{(1)} \tan(\alpha_1) + Z_{(2)} \tan(\alpha_2) \right) +$$
$$\left[ \left( Z_{(2)}^2 - Z_{(1)}^2 \right) \tan(\alpha_1) \tan(\alpha_2) \right]^2$$

and:

$$\alpha_j = k_{z(j)} d_j = \pm (2\pi/\lambda) n_j d_j \sqrt{1 - (\rho/n_j)^2}.$$

### 2.4. Band Gap Maximum Extinction Per Length

As a rule, in practical applications, we have values of two RIs ($n_1$ and $n_2$) of alternative media in the 1D PC, and the purpose is to find the thickness of each alternative layer, which provides the maximum extinction (or the maximum extinction per length) at given RIs, wavelength and angle. The first method for maximizing extinction at pre-selected wavelength and wavevector is to use 'quarter-wave-length' layer's thicknesses for the double layer, which are:

$$d_j = \frac{\lambda}{4n_j \cos(\theta_j)} = \frac{\lambda}{4\sqrt{n_j^2 - \rho^2}}, \tag{8}$$

where $j = 1$ or 2.

The second method is more optimal, especially at large incident (grazing) angles (i.e., at $\cos(\theta_j) \to 0$), and it permits one to find the desired values of the thicknesses $d_1 = d_{1\text{max}}$ and $d_2 = d_{2\text{max}}$ which maximize expression (20) in [36] and provide the maximum extinction per length $d_1 + d_2$. In the first step of the program, a user may choose either of these two methods to determine the double layer thicknesses ($d_1$ and $d_2$).

### 2.5. Dispersion Relation for PC SM and Its Solution for the Truncated Layer Thickness

There are several programs and algorithms for calculating electromagnetic wave propagation through planar stratified media [40,41]. In contrast to them, our program additionally calculates the thickness of the truncated layer $d_3$ making propagation of PC SM possible. The last (or penultimate—see below) layer with impedance $Z_{(3)}$ must be truncated to ensure the propagation of surface waves along the planar interface of stratified media at a given wavelength and wavevector.

A general condition for the existence of a surface mode between two media with impedances $Z_{\text{left}}$ and $Z_{\text{right}}$ is:

$$Z_{\text{left}} + Z_{\text{right}} = 0. \tag{9}$$

In our case, this condition takes the form:

$$Z^{\text{into}}_{(PC\&Z_{(3)})} + Z_{(e)} = 0. \tag{10}$$

The input impedance of a 1D PC structure plus the truncated layer, according to (3), will be:

$$Z^{\text{into}}_{(PC\&Z_{(3)})} = Z_{(3)} \frac{Z^{\text{into}}_{(PC)} - iZ_{(3)} \tan(\alpha_3)}{Z_{(3)} - iZ^{\text{into}}_{(PC)} \tan(\alpha_3)}. \tag{11}$$

By solving Equations (10) and (11), we obtain the dispersion relation for the optical surface waves in the 1D PC:

$$\alpha_3 \equiv \left[ k_{z(3)} d_3 \right] = \pi M + \arctan\left( \frac{-i\left( Z^{\text{into}}_{(PC)} + Z_{(e)} \right) Z_{(3)}}{Z^2_{(3)} + Z^{\text{into}}_{(PC)} Z_{(e)}} \right), \tag{12}$$

where $M$ is a whole number.

If the last dielectric layer to be truncated is followed by a thin metal layer, sustaining long-range surface plasmons [42], then one can use Equation (12), where the external medium impedance $Z_{(e)}$ is replaced by the input impedance of the external medium plus an impedance of the metal films with a pre-selected thickness $d_{\text{m}}$, using (3): $Z^{\text{into}}_{(e)\&(\text{m})}$. In this case, the thickness of the last dielectric layer (or the penultimate layer, if we are counting the metal layer) can ensure the propagation of long-range surface plasmons along the external surface if the condition for minimizing the electromagnetic field inside the metal film is satisfied. In our program, this conditions (see (31) in [36]):

$$\rho_{1/2} = n_e + \frac{n_e^3}{2} \left[ \pi \frac{d_{\text{m}}}{\lambda} \right]^2, \tag{13}$$

can be selected by the appropriate checkbox.

## 3. Results

### 3.1. Practical Implementation in the Program

An implementation of the impedance approach described in this paper is available as a free Windows program at: https://www.pcbiosensors.com/1DPC4all.htm (accessed on 1 October 2022). Both the program interface and numerical calculations were implemented in the C# programming language within .NET 6 framework. The program is distributed as a self-contained single file, which contains all components of the application, including the .NET libraries and target runtime libraries. The program is isolated from other .NET applications and does not use a locally installed shared runtime. The executable file `1DPC4all.exe` can be run on any 64-bit Windows above Windows 7. The user of the program is not required to download and install any versions of .NET.

### 3.2. Refractive Indices Data

The refractive indices data of the layers and their dispersion can be represented in the form of the Sellmeier formulas for dielectric layers and the Drude formulas for metal layers. Alternatively, the RI dispersion can be presented as a set of experimental data in the form of a wavelength-n-k table. Users can add their RI data to the existing ones in all of these forms.

#### 3.2.1. Sellmeier Formula

The seven coefficients for the Sellmeier formula may be written in an ASCII file with the extension `.slmr` in subfolder '\1DPC4all\Resources\calcRI\'. These coefficients ($c_i$) will be substituted into the Sellmeier formula in the following form:

$$n^2 = c_0 + \frac{c_1 \lambda^2}{\lambda^2 - c_2} + \frac{c_3 \lambda^2}{\lambda^2 - c_4} + \frac{c_5 \lambda^2}{\lambda^2 - c_6}, \tag{14}$$

where $\lambda$ is in μm. As an example, the program offers the following coefficients for BK7 glass: $c_{[0 \to 6]}$=[1 1.04 0.006 0.23 0.02 1.01 103.56] in the file '\1DPC4all\Resources\calcRI\BK7.slmr' (which can be changed by users, if desired).

### 3.2.2. Drude Formula

The coefficients for the Drude formula may be also written in subfolder '\1DPC4all\Resources\calcRI\' as an ASCII file with the extension .drd. These nine coefficients ($c_i$) will be substituted into the Drude formula in the following form:

$$n^2 = c_0 - \frac{c_1^2}{\omega^2 + i\omega c_2} + \frac{c_3^2}{c_4^2 - \omega^2 - i\omega c_5} + \frac{c_6^2}{c_7^2 - \omega^2 - i\omega c_8}, \qquad (15)$$

where $\omega = 1/\lambda$ in 1/μm. As an example, the program offers the following coefficients for metallic silver: $c_{[0 \to 8]}$=[1.881 7.0235 0.05494 1.9591 3.6032 0.44315 5.2542 4.6022 1.8702] in the file '\1DPC4all\Resources\calcRI\Ag.drd' (users can change it, if desired).

In both cases, the user can insert $c_i = 0$ for the trailing $i$ if the shorter form of the Sellmeier/Drude equation is used.

### 3.2.3. Experimental n-k Dataset

Users can also represent the real and imaginary parts of RI as a tab/space-separated table, where each line has the form '$\lambda$[Å] n k'. This table must be saved as an ASCII file with the extension .nk and added to the zip file '\1DPC4all\Resources\allRI.zip', where 188 .nk files (representing various materials) are already stored. The values of n and k between the wavelengths presented in the table will be found by linear interpolation.

### 3.2.4. Maxwell Garnett Approximation for Mixed Layers

The effective medium theory, in the form of the Maxwell Garnett approximation [43], can also be used for RI of mixed layers. The RI of a mixed layer is specified in an ASCII file with the extension .gnt in the subfolder '\1DPC4all\Resources\calcRI\'. Each line in the file must represent the matrix medium and one (or two) inclusion(s) followed by their volume percentage. For example, a file with lines: 'SiO2.nk 97   Au.drd 2   air.slmr 1 ' will provide an effective RI for a layer consisting of 97% $SiO_2$ as a matrix material (given as .nk dataset '$\lambda$ n k'), 2% Au inclusion (represented by the Drude formula) and 1% of air bubbles (represented by Sellmeier formula). Such file can also be automatically created from two or three selected layers in Step 2 of the program, where the volume percentage will be given by the thicknesses of the layers (hints in the program will provide all the details).

Note that the Maxwell Garnett theory gives a good approximation when one of the materials (the matrix) prevails over the inclusions in terms of volume percentage.

### *3.3. Step 1: Selection of Double Layer Materials, Wavelength and Angle*

The design of 1D PC structures in the program is divided into three simple steps. After starting the program, the window of Step 1 appears (see Figure 3). In this window, the user must select the double layer materials ($n_1$ and $n_2$) and set the (central) wavelength and angle of incidence of the optical waves. If the angle of incidence is greater than the TIR angle ($\rho > n_0$), the program will also calculate the final layer thickness $d_3$ (with RI $n_3$) that sustains the surface optical modes. But if the user sets $\rho < n_0$ (no TIR), the program will not calculate the final layer (sustaining surface optical modes), and in the following steps it will only be possible to calculate reflection and transmission through 1D PC (without PC SM).

After selecting the quarter-wavelength thickness (or the optimal thickness) checkbox, the button to Step 2 will be enabled, as indicated by the green dotted arrows in Figure 3. The program will simultaneously calculate the thicknesses of the double layers $d_1$, $d_2$ and the possible thicknesses of the third layer, $d_3(M)$, which will sustain surface waves at a given wavelength and at a given angle.

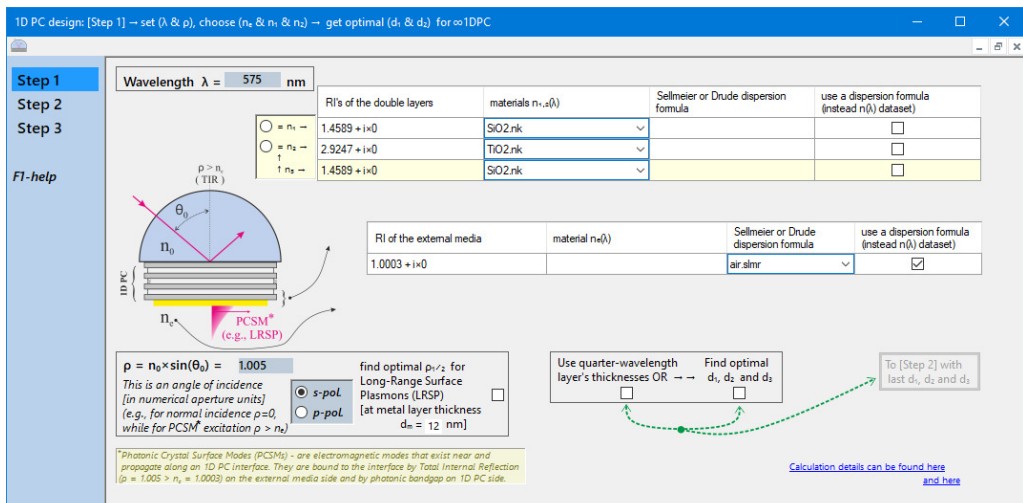

**Figure 3.** Step 1 program's window for selection of double layer materials, wavelength and angle.

### 3.4. Step 2: Choosing the Number of Layers and Final Adjustment of the Structure

In the Step 2 window shown in Figure 4, the user must select the number of layers for the 1D PC structure. In the Kretschmann scheme, the optimal number of layers (for given losses in the structure) can be checked by the resonant dip of the reflection coefficient, which should drop to zero when the number is optimal. Also, at this step, any final adjustments to thicknesses, RIs, and materials of structure may be made.

**Figure 4.** Step 2 program's window for choosing the number of layers and final adjustment of the structure.

### 3.5. Step 3: Presentation and Analysis

At Step 3 of the program, one can visualize the intensity reflectance ($R = |r|^2$) and transmission ($T = |t|^2$) coefficients of the structure as a function of the wavelength $\lambda$ or the angle $\rho$ (see Figure 5). Note that in the case of TIR, $T$ describes the enhancement of the evanescent field intensity at zero distance from the interface due to surface waves excitation. In this case, the power transmission in far-field is zero ($T = 0$ at $z \gg \lambda$). Thus, the usual condition for the energy flow through the interface:

$$|r|^2 + \frac{n_e \cos(\theta_e)}{n_0 \cos(\theta_0)} |t|^2 = 1 \tag{16}$$

is satisfied in the far zone.

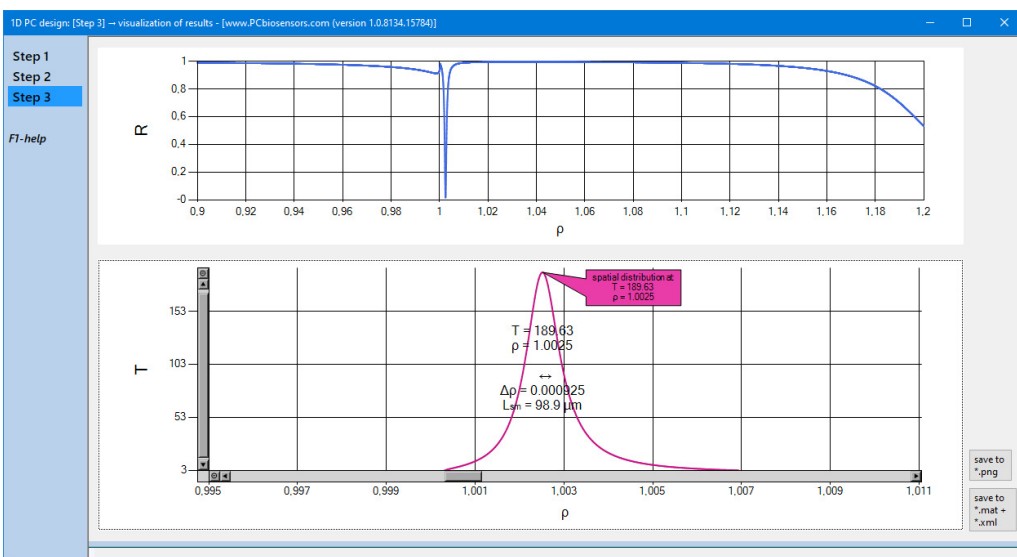

**Figure 5.** Reflection and transmission in Step 3 program's window.

It is also possible to choose any point of the angular dependence $T(\rho)$ to visualize the spatial distribution of the field (see Figure 6) at the selected angle $\rho$. One can see in Figure 6 that the electromagnetic fields reach their minimum in the center of the metal film (layer #14), which minimizes total losses in the metal film when the condition (13) is satisfied. This condition for $\rho_{1/2}$ can be selected in the Step 1 by an appropriate checkbox. The dispersion of the structure can also be calculated and visualized in the form of transmission field enhancement near the interface $\log_{10} T(\lambda, \rho)$, as shown in Figure 7. It should be noted that the program considers the incident wave as a plane wave (i.e., as an infinite flat front) and makes all calculations under this assumption. If the incident wave is constrained (e.g., focused to a size smaller than the propagation length of PC SMs), then additional fine interference structures may appear on experimental reflection curves, as described in [44].

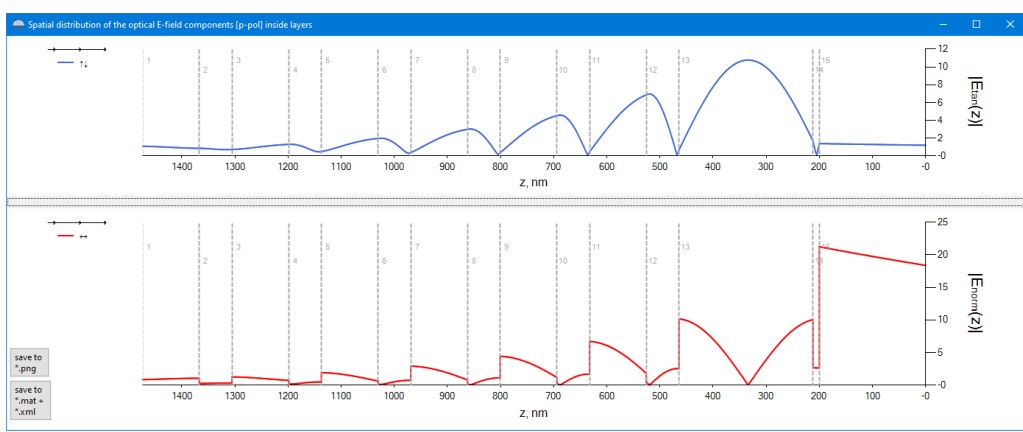

**Figure 6.** The spatial field distribution inside 1D PC.

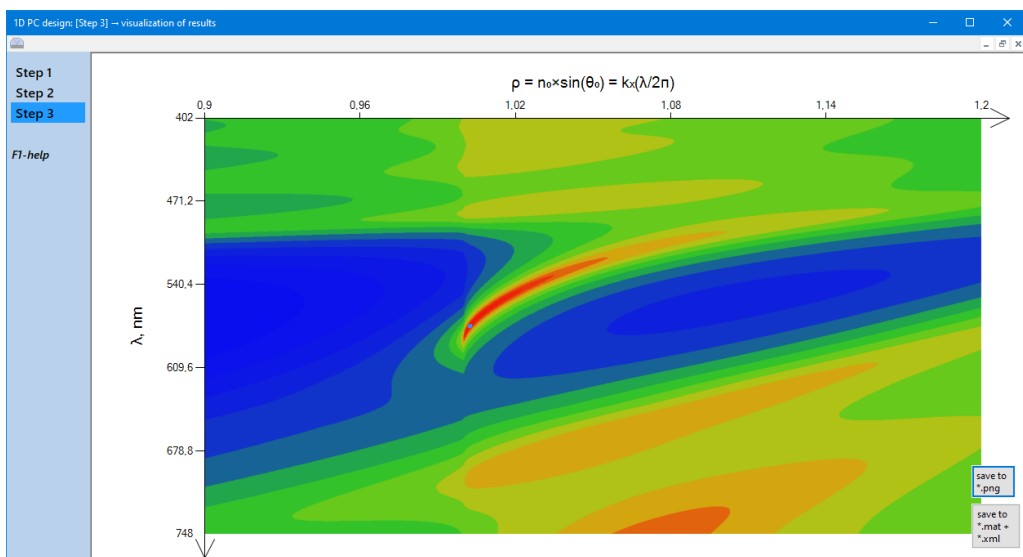

**Figure 7.** Optical dispersion in Step 3 program's window.

### 3.6. Additional Features

The program also facilitates the calculation of some special 1D PC and their features.

### 3.6.1. 1D PC Structures with Two Metal Nanolayers

It has recently been shown [29,37], that it is possible to design multilayer systems, in which *two* metal nanolayers—one on each side of a thin dielectric film—sustain the long-range propagation of SPs (LRSPs). It this case, the correct thickness of the metal-bounded dielectric film is important, which ensures that the optical electric field has minima inside *both* metal nanolayers, and LRSP propagation will be possible in structures containing *two* metal nanolayers.

In Step 2, the user can calculate and insert a dielectric film of the proper thickness, sandwiched between two metal layers, into a structure sustaining LRSP (if the checkbox for $\rho_{1/2}$ was preliminary selected in the Step 1). An example of such a structure taken from [29] is presented in the '\1DPC4all\savedResults\NanoMicroLetters2020\' subfolder of the program.

### 3.6.2. Luminescence from 1D PC structures

One of the possible applications of such structures is the electroluminescence of an active (sandwiched) film in PC [29]. The program can calculate the dispersion of luminescence from layers that RI values in Step 2 marked with '+−', '−−' or '++'. These calculations are performed using the reciprocity theorem [45,46]. In this case, it means that the integral optical electric field generated in the far zone by the dipoles located in the luminescence layer is the same as the electric field from the far zone dipoles (i.e. from plane optical waves) generated in this (marked) layer.

### 3.7. A Practical Example of Designing a 1D PC Structure Sustaining Long-Range Surface Plasmons

A step-by-step illustration of the implementation of the program for designing a multilayer structure, sustaining long-range surface plasmons, can be seen in Figures 3–7. Suppose we have $SiO_2$ and $TiO_2$ as materials for the double layer, a laser with $\lambda = 575$ nm as the light source, and we want to get long-range surface plasmon propagation on a 12 nm-thick gold film at this wavelength. At Step 1 (see Figure 3) we select *SiO2.nk* as $n_1$, *TiO2.nk* as $n_2$ in the double layer table, and *SiO2.nk* as $n_3$ for the penultimate layer (before the final layer *Au.drd* as $n_m$).

Next, we set $d_m = 12$ nm and check the box nearby to "*find the optimal $\rho_{1/2}$…*". The polarization radio-button automatically changed to "*p-pol.*" (only the p-polarized wave can excite surface plasmons) and an angle (optimal for long-range propagation)

$\rho = \rho_{1/2} = 1.00245$ will be indicated in the corresponding text field. Then we can find the optimal double layer thicknesses for the selected materials and the specified $\lambda$ and $\rho$. After checking the box "*Find optimal $d_1$, $d_2$ and $d_3$*" the program will calculate and display these optimal thicknesses $d_1 = 106.86$ nm, $d_2 = 61.42$ nm and $d_3$(M = 1) = 252.16 nm (calculated for semi-infinite 1D PC) and the button "*To Step 2 with last $d_1$, $d_2$ and $d_3$*" becomes active.

After pressing this button, we proceed to Step 2 (see Figure 4) where we should choose an appropriate number of layers for our 1D PC structure. The easiest way to do this is to change the number of layers up and down and check the reflection profile $R(\rho)$ for each number by pressing the button "*To Step 3 to plot data . . .*". When a plasmon resonance dip drops to zero ($R(\rho_{res}) \sim 0$), the number of layers is optimal for the structure with the current level of internal loss. For the structure under consideration, this optimal number is $N = 14$. So, the design of the structure, sustaining long-range surface plasmons, is completed and parameters, highlighted by bold font above, may be transferred to a coating service. In the following Step 3 (see Figure 5), we can analyze the output parameters of the structure, such as field enhancement, propagation length of surface waves, and so on.

For example, analyzing the spatial distribution of the field, one can be convinced that the total losses in the structure are minimal, since the electromagnetic fields reach their minimum inside the metal film (see Figure 6). Also, by changing the radio-button to "*Dispersion . . .*" and by pressing button "*To Step 3 to plot data . . .*", we can calculate and visualize the dispersion of the structure and look at a dispersion curve of the long-range surface plasmons (see Figure 7). Returning to the previous steps, one can introduce various alteration to the structure and look how this will affect the output parameters of the structure.

## 4. Discussion

We have presented a software for calculations of one dimensional PC structures. To the best of our knowledge, this is the first free program that can not only calculate the reflection and transmission of optical waves through a multilayer coating, but also calculate parameters for the excitation of surface optical waves propagating along the interface. The use of the impedance approach made it possible to develop software that calculates the thickness of the last (or penultimate) layer, as well as the thickness of the active layer (sandwiched between two metal nanolayers) that sustain surface waves. We hope that this program should facilitate scientific development and implementation of practical applications in this growing field.

**Funding:** This research was funded by the Russian Science Foundation grant 22-22-00836.

**Data Availability Statement:** The Windows program described in this article can be downloaded from: https://www.pcbiosensors.com/1DPC4all.htm (accessed on 1 October 2022) .

**Conflicts of Interest:** The author declares no conflict of interest.

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
