# Peer review of "Design of 1D Photonic Crystals Sustaining Optical Surface Modes"

_coatings, doi:10.3390/coatings12101489_

Round 1

Reviewer 1 Report

The authors presented a software for calculations of one-dimensional PC structures. Technically, the authors employ a free Windows program based on an impedance approach with apparent authority, and the results are interesting for the readers. However, the necessary elucidation of the mechanism and references are absent in the manuscript to explain the obtained results. Therefore, in my opinion, a major revision is needed to accommodate the high-quality requirements of this Journal. 

1.    The incident light pattern used in the simulations should be mentioned in the text.

2.   It states, “Such materials can exhibit photonic band gaps that are very much like the electronic band gaps for electron waves traveling in the periodic potential of the crystal.” A related reference (e.g., Opt. Express, 2011, 19(6), 4862-4872) should be added in the text. Besides, to be beneficial for the readers to know the properties of photonic crystal, e.g., high birefrigence (see Prog. Electromagn. Res. B, 2010, 22, 39-52 and J. Appl. Phys., 2011, 109, 093103), several literatures regarding the features of photonic crystals should be included in the introduction section.

3.   Page 5, the seven coefficients for Sellmeier equation should be mentioned in the text or cite a related reference, and in the same manner to the Eq. (15).

4.      Can this software calculate the bandgap of proposed structure? If possible, please show the bandgap of the proposed layer material in the revised manuscript.

5.  The expression of T in Fig. 5 should be defined in the text. Is it transmittance? Why T value exceeds 1 in Fig. 5?

6.      The results obtained from Fig. 5 should be described in more detail (e.g., the relationship between R and T).

7.      Page 8, it states “it is possible to design multilayer systems in which two metal nanolayers”. If possible, can you include one example of metal nanolayer in the revised manuscript.

Reviewer 2 Report

The authors provided a free windows program for designing the structures of one-dimensional photonic crystals. One-dimensional photonic crystals can have surface states, and the dispersion curve of the surface states can be adjusted by adjusting the thickness of the layer, so that the surface wave mode to be excited can be designed flexibly and conveniently. Because of the strong surface electric field and its sensitivity to surface changes, one-dimensional photonic crystal surface wave sensors can be developed. The program designed by the authors can theoretically design a one-dimensional photonic crystal Bloch surface wave sensor. The influence of the structure layer of photonic crystal and the physical properties of the medium composing the photonic crystal is discussed by using the transmission characteristics such as field distribution. So that it can play a guiding role in the preparation and application of one-dimensional photonic crystals. Importantly, the program is free, and developers can download and use it freely. I think the work is good and can help more researchers in this field. I can recommend that it be published in Coating.

Reviewer 3 Report

A software for calculations of reflection and transmission through a multilayer coating in a  one-dimensional PC structure is reported. Parameters for the excitation of surface optical waves propagating along the interface are determined as well.

I think the paper is scientifically sound. Novelty could be better underline, but my overall evaluation is positive.

Round 2

Reviewer 1 Report

The author has addressed my comments and I suggest accepting this manuscript.